# Microtransplantation of Postmortem Native Synaptic mGluRs Receptors into *Xenopus* Oocytes for Their Functional Analysis

**DOI:** 10.3390/membranes12100931

**Published:** 2022-09-26

**Authors:** Brice Miller, Naomi Moreno, Berenice A. Gutierrez, Agenor Limon

**Affiliations:** Mitchell Center for Neurodegenerative Diseases, Department of Neurology, The University of Texas Medical Branch, Galveston, TX 77555, USA

**Keywords:** G-protein coupled receptors, post-mortem, phospholipase C, mGluR, microtransplantation of synaptic membranes

## Abstract

Metabotropic glutamate receptors (mGluRs) are membrane receptors that play a central role in the modulation of synaptic transmission and neuronal excitability and whose dysregulation is implicated in diverse neurological disorders. Most current understanding about the electrophysiological properties of such receptors has been determined using recombinant proteins. However, recombinant receptors do not necessarily recapitulate the properties of native receptors due to the lack of obligated accessory proteins, some of which are differentially expressed as function of developmental stage and brain region. To overcome this limitation, we sought to microtransplant entire synaptosome membranes from frozen rat cortex into *Xenopus* oocytes, and directly analyze the responses elicited by native mGluRs. We recorded ion currents elicited by 1 mM glutamate using two electrodes voltage clamp. Glutamate produced a fast ionotropic response (6 ± 0.3 nA) in all microtransplanted oocytes (*n* = 218 oocytes) and a delayed oscillatory response (52 ± 7 nA) in 73% of them. The participation of Group 1 mGluRs was confirmed by the presence of metabotropic oscillations during the administration of (±)-1-Aminocyclopentane-trans-1,3-dicarboxylic acid (ACPD; Group 1 mGluR agonist), and the absence of oscillations during co-administration of N-(1-adamantyl)quinoxaline-2-carboxamide (NPS 2390; Group 1 mGluR antagonist). Since both mGluR1 and mGluR5 belong to Group 1 mGluRs, further investigation revealed that mGluR1 antagonism with LY 456236 has little effect on metabotropic oscillations, while mGluR5 antagonism with 100 µM AZD 9272 has significant reduction of metabotropic currents elicited by ACPD and glutamate. We confirmed the expression of mGluR1 and mGluR5 in native synaptosomes by immunoblots, both of which are enhanced when compared to their counterpart proteins in rat cortex tissue lysates. Finally, these results demonstrate the merit of using microtransplantation of native synaptosomes for the study of mGluRs and the contribution of mGluR5 to the metabotropic glutamate signaling, providing a better tool for the understanding of the role of these receptors in neurological disorders.

## 1. Introduction

Glutamate is the major excitatory neurotransmitter of the central nervous system, driving the neuronal communication and excitability that underlies major brain processes [1]. The neurochemical effects of glutamate are carried out by receptors localized in the cellular membrane [2]. These receptors can be categorized as ionotropic and metabotropic according to their structure, activation and signaling [3]. Ionotropic receptors are fast responding ligand-gated ion channels that are subdivided into four groups: NMDA, AMPA, kainate and delta receptors [4,5]. On the other hand, metabotropic receptors are G-protein coupled receptors which are subdivided into Group I, Group II, and Group III receptors, depending on their specific signaling [6]. Group I includes the metabotropic glutamate receptors 1 (mGluR1) and 5 (mGluR5), both of which utilize the G_αq_ pathway and mediate inositol trisphosphate-induced Ca^2+^ mobilization [7]. Group II consists of mGluR2 and mGluR3 and use the G_i/o_ pathway [8,9]. Group III includes mGluR4, mGluR6, mGluR7 and mGluR8 and similarly use the G_i/o_ pathway [8,9]. Importantly, altered glutamatergic pathways have been observed in Alzheimer’s Disease, epilepsy and major depressive disorder [10,11,12]. For example, mGluR1 reduces amyloid-beta production by controlling the cleavage of amyloid precursor protein, a crucial regulatory step in the development of Alzheimer’s Disease [13], although findings on how mGluR1 expression changes throughout the course of the disease are conflicting [14,15,16]. Contrary to mGluR1, mGluR5 appears to aid in amyloid beta-dependent synaptotoxicity through interactions with cellular prion protein [17,18]. Negative modulation of mGluR5 reduces cognitive decline associated with Alzheimer’s Disease, suggesting that mGluR5 may be upregulated or overactive in the disease [19,20]. In addition to interactions with prion protein, mGluR5 activity is strongly regulated by its interactions with scaffolding proteins like the Na(+)/H(+) exchanger regulatory factor, NHERF-2 [21], as well as postsynaptic scaffolding proteins Homer and Shank, which mediates the clustering of mGluR5, highlighting a complex regulatory mechanism of mGluRs activity [22,23,24,25,26]. However, a major roadblock in understanding the properties of metabotropic receptors in human disease is the limited access to study their pharmacological characteristics directly within the membrane complexes where they are integrated. Recombinant expression of metabotropic receptors do not necessarily reproduce the function of the native receptors [27,28,29]. Here, we used the microtransplantation method, which was originally described by Miledi’s group in 1995 to microtransplant nicotinic receptors from torpedo electroplaques into *Xenopus* oocytes [30], and later used for the characterization of brain cortex GABA and AMPA ionotropic receptors in humans. Briefly, proteoliposomes containing native receptors within native cellular membranes are injected into *Xenopus* oocytes wherein they fuse with the oocyte membranes providing a living environment for the transplanted receptors (please see Eusebi et al., 2009 and Zwart et al., 2019 [31,32] for a comprehensive historical review of the method). For this study we took advantage of accumulated proteomic data demonstrating that scaffolding proteins like Homer and Shank are consistently found in synaptosomal preparations [33,34]; therefore, we used the microtransplantation of synaptic membranes (MSM) from postmortem frozen rat cortex into *Xenopus* oocytes in an effort to preserve a more physiological structural relationship between mGluRs and their interacting proteins, thus potentially aiding a better understanding of the role of these receptors in diverse neurological disorders.

## 2. Materials and Methods

### 2.1. Xenopus Oocytes

Oocytes were extracted from *Xenopus laevis* frogs in accordance with the National Institutes of Health Guide for the Care and Use of Laboratory Animals, and the Institutional Animal Care and Use Committee (IACUC) at The University of Texas Medical Branch at Galveston (IACUC:1803024), as reported before [34]. Briefly, frogs were anesthetized by immersion in 0.17% MS-222 bath, euthanized and the ovaries were surgically removed. Oocytes were isolated and defolliculated by gentle rotation in Barth’s solution (88 mM NaCl, 1 mM KCl, 0.41 mM CaCl_2_, 0.82 mM MgSO_4_, 2.4 mM NaHCO_3_, 5 mM HEPES [pH 7.4 with NaOH]) to which was added 2 mg/mL collagenase (Sigma, Sant Louis, MO, USA) at 30 °C for 4 h. Healthy-looking oocytes were manually selected and stored at 16 °C in fresh Barth’s until injection of synaptic membranes.

### 2.2. Brain Samples

Brain cortex and cerebellum were isolated from adult Wistar rats, 2 months old. Animals were euthanized in accordance with the National Institutes of Health Guide for the Care and Use of Laboratory Animals, and under the IACUC at the University of California Irvine (IACUC: 1998-1388). Freshly dissected brains were surgically removed and snap-frozen by immersion in liquid nitrogen and stored at −80 °C.

### 2.3. Synaptosome Isolation

Synaptosome enriched preparations were isolated from 50 mg of brain cortex from four rats using the Syn-PER reagent protocol (Thermofisher, Waltham, MA, USA). Briefly, 50 mg of cortex from each rat was homogenized in Syn-PER reagent to which was added an EDTA free protease inhibitor (Thermofisher, Cat#. A32955). Synaptosomes were enriched in the P2 fraction after the last centrifugation, with P1 and S1 fractions enriched in myelin, nuclei, and any non-homogenate tissue or cytosolic elements. Resulting P2 proteins were re-suspended in Syn-PER and quantified via fluorometry using a DeNovix QFX unit (DeNovix, New Castle, DE, USA) and flowcytometry using a Guava EasyCyte (Guava Soft v2.7, Luminex, Austin, TX, USA) [35]. For the preparation of cerebellar synaptosome enriched preparations, 50 mg of the whole cerebellum was used following the same procedure described above.

### 2.4. Injection of Synaptosome Preparations

50 nL of P2 fraction, diluted to 2 mg/mL in filtered deionized water, was sonicated three times for 1 min using an FS20D ultrasonic cleaner (Thermofisher) and microinjected using a Nanoject II (Drummond) into the pigmented animal side of the *Xenopus* oocytes, just beneath the membrane, avoiding the nucleus [36].

### 2.5. Two Electrode Voltage Clamp

Recordings were performed by perfusing (5–10 mL/min) microtransplanted oocytes with Ringer’s solution (115 mM NaCl, 2 mM KCl, 1.8 mM CaCl_2_, 5 mM Hepes [pH 7.4 with NaOH]) in a chamber (≈0.1 mL) at room temperature (19–21 °C) and measured with microelectrodes containing 3 M KCl with resistance ranging between 0.5 to 3 MΩ. Oocytes were voltage clamped at -80 mV using an Oocyte Clamp OC-725C amplifier (Warner Instruments, Holliston, MA, USA). Observed ion currents were filtered by Dual variable filter Kemo at 10 Hz and recorded with WinEDR version 3.9.1 Strathclyde Electrophysiology Software (John Dempster, Glasgow, UK). Recordings took place 1–3 days after injection.

### 2.6. Drugs

1-Aminocyclopentane-trans-1,3-dicarboxylic (ACPD), *N*-(1-adamantyl)quinoxaline-2-carboxamide (NPS 2390), 6-Methoxy-*N*-(4-methoxyphenyl)-4-quinazolinamine hydrochloride (LY 456236) and 3-Fluoro-5-[3-(5-fluoro-2-pyridinyl)-1,2,4-oxadiazol-5-yl]benzonitrile (AZD 9272) were obtained from Tocris (Minneapolis, MN, USA); GABA was from Sigma-Aldrich (St. Louis, MO, USA). Concentrated stocks of NPS 2390, LY 456236 and AZD 9272, were prepared in DMSO and stocks of ACPD, glutamate and GABA were prepared in filtered deionized water. All stock solutions were stored as aliquots at −20 °C. Experimental dilutions were prepared the day of recording in Ringer’s solution.

### 2.7. Immunoblots

For immunoblots, brain cortex synaptosomes, cortical and cerebellar homogenates samples were briefly sonicated and lysed with RIPA 1X buffer. The rat lysates were run under non-denaturing conditions for mGluR1α and denaturing conditions for mGluR5 on Bolt-Bis-Tris plus gels 4–12% and transferred to nitrocellulose membranes. The samples were probed with monoclonal mouse antibodies against mGluR1α (1:1000; BD Pharmingen, San Diego, CA, USA), rabbit monoclonal antibodies against mGluR5 (2:10,000; Abcam, Cambridge, UK) and polyclonal rabbit antibodies against β3-tubulin (1:10,000; loading control; Abcam).

### 2.8. Statistics

The data presented in this work were obtained from four different rat brain cortex and one cerebellar synaptosomal preparations. All analyses were performed using JMP Pro 16 (SAS, Cary, NC, USA) and Prism Graphpad software package (GraphPad Software, San Diego, CA, USA).

## 3. Results

### 3.1. Oscillatory Responses to Glutamate in Oocytes Microtransplanted with Cortical Synaptic Membranes

This study focuses on the characterization of synaptosome enriched preparations from the brain cortex. For this, we first compared the responses elicited by 1 mM glutamate on microtransplanted oocytes after 24 h of injection with cortex synaptosomes and naïve non-injected oocytes. Non-injected oocytes do not express endogenous glutamate receptors [37,38], consequently, they did not show responses to glutamate perfusion (Figure 1A). Conversely, all microtransplanted oocytes showed a fast activating and sustained current of 6 nA ± 0.3 nA (*n* = 218 oocytes) when perfused with glutamate (Figure 1B). This sustained current had a similar profile to ion currents elicited by AMPA receptors in previous studies [37]. Microtransplanted oocytes also showed an oscillatory response of 52.2 ± 7.3 nA (*n* = 159 oocytes) which appeared after about 11 s (range 3 to 38 s) of continued exposure to glutamate (Figure 1B,C). Importantly, this oscillatory response was significantly reduced after 48 h of the initial injection, from 62 ± 9.7 nA (*n* = 117) at 24 h to 26.1 ± 4.4 nA at 48 h (*n* = 42; *p* = 0.03; two-tailed *t* test). Only 19% (*n* = 42) of microtransplanted oocytes showed this response at 48 h. Conversely, the fast-activating response did not show significant changes after 48 h (5.8 ± 0.4 nA at 24 h; *n* = 117 vs. 6 ± 0.6 nA at 48 h; *n* = 42; *p* = 0.82) (Figure 1C). Small, if any, oscillatory responses were observed in oocytes injected 11 days after oocytes were extracted from the ovaries (5 ± 3.4 nA; *n* = 2 oocytes out of 6 tested), this despite a strong fast-activating ionotropic response that was observed in the same oocytes (14.5 ± 3.5; *n* = 6) indicating a successful fusion of synaptic membranes but a failure of mGluR to couple with the intracellular machinery of the oocyte. As an additional control, we also tested GABA responses in all oocytes to ensure successful insertion of membranes when glutamate responses were small. The reduction of oscillatory responses indicates a time dependence on their capacity of activation or in their successful coupling with the intracellular signaling of the oocyte.

### 3.2. Participation of Cortical Group I Metabotropic Glutamate Receptors in Oscillatory Responses

*Xenopus* oocytes are known to produce membrane oscillations upon activation of endogenous phospholipase C which in turn activates a large endogenous Ca^2+^-dependent chloride current [39,40,41,42]; therefore, we hypothesized that oscillatory responses in microtransplanted oocytes were carried out by Group I mGluRs which also activate phospholipase C-mediated signaling inside the oocyte. To test this hypothesis, we perfused microtransplanted oocytes with ACPD, which is an agonist of Group I and Group II mGluRs [43]. Because activation of recombinantly expressed Group II mGluRs does not elicit oscillatory currents, as it is characteristic of G-protein coupled receptors that inhibit adenylyl cyclase [44], ACPD is expected to only activate Group I mGluRs. Here we found that 100 μM ACPD elicited small, if any, fast responses of 0.5 ± 0.3 nA (*n* = 39 oocytes) but elicited oscillatory responses of 18.1 ± 2.6 nA (*n* = 39 oocytes) (Figure 2A). Further analysis showed that ACPD elicits oscillatory responses in a concentration dependent manner with an EC_50_ of 94.4 μM (Figure 2B).

To confirm the participation of Group I mGluRs, we tested NPS 2390, which is a specific antagonist for Group I mGluRs. As expected, 10 μM NPS 2390 completely inhibited oscillatory responses elicited by glutamate or ACPD (Figure 3A–D). Notably, the effect of NPS 2390 was not reverted even after 20 min of washing and recovery. These results confirm the presence of functional Group I mGluRs in synaptic membranes and their successful coupling to the inositol triphosphate signaling in the oocyte by microtransplantation methods.

### 3.3. Cortical mGluR 5 Type Receptors Are Mostly Responsible for Oscillatory Responses

To determine how much mGluR1 and mGluR5 each contributed to metabotropic oscillations, selective antagonists were used to test their effects on responses elicited by ACPD and glutamate. LY 456236, an mGluR1 selective antagonist, was tested at a 1 µM concentration on responses elicited by glutamate (*n* = 13 oocytes) or ACPD (*n* = 6 oocytes). LY 456236 did not inhibit the oscillatory responses by either agonist, suggesting that mGluR1 does not significantly participate in generating the oscillatory responses (Figure 4A–D).

We next looked at the contributions of mGluR5 on metabotropic responses. Blocking of mGluR5 was evaluated using AZD 9272, a mGluR5 specific antagonist, at 1 µM. Mean glutamate responses were 4.9 ± 1.1 nA (*n* = 9 oocytes) for ionotropic and 39.1 ± 23.1 nA (*n* = 9 oocytes) for metabotropic (Figure 5A). The data showed significant reduction of oscillatory current amplitude confirming mGluR5 participation in metabotropic responses. As for ACPD, no fast ionotropic responses were observed and metabotropic had a mean response of 12.7 ± 5.4 nA (*n* = 9 oocytes) (Figure 5B).

### 3.4. Presence of mGluR1 and mGLuR5 in Cortical Synaptosomal Preparations

Immunoblots confirmed the presence of mGluR1α in rat synaptosomes and cortex homogenate. We also included cerebellum homogenate for comparison. Protein expression of mGluR1α in synaptosomes was similar to cerebellum, a region known to be enriched for this protein [45]. Moreover, mGluR5 showed a ∼6-fold increase in synaptosomes compared to cortex, from 0.25 to 1.62 A.U., measured by duplicate. Both mGluRs bands were found at ∼150 kDa, close to the predicted molecular weight of 132 kDa (Figure 6A,B). Interestingly, oocytes microtransplanted with synaptosome preparations isolated from the cerebellum showed a fast response of 2.31 ± 0.51 nA (*n* = 9 oocytes) and an oscillatory response of 35.74 ± 8.38 nA (*n* = 9 oocytes) (Figure 6C,D). Further characterization of the cerebellar oscillatory response will be done in future studies.

## 4. Discussion

Our results show that synaptosomes isolated from frozen cortical rat tissue and microtransplanted into the plasma membranes of *Xenopus* oocytes contain metabotropic glutamate receptors that can couple with intracellular signaling of the oocyte. This opens the possibility for detailed pharmacological characterization of native mGluRs from postmortem rodents and potentially of human samples. Using MSM rather than heterologous methods to study native synaptic mGluRs also allows for preservation of the structural relationships of these receptors with other interacting proteins [46]. Group I mGluRs are preferentially expressed at postsynaptic terminals where they likely undergo crosstalk with ionotropic glutamate receptors directly, as well as through scaffold-mediated interactions with proteins such as Homer, Shank and PSD-95 [26]. The study of mGluR complexes through MSM means that our data may reflect native interactions between mGluRs and ionotropic glutamate receptors. Future characterization of those relationships in synaptosomes, as well as the modulatory effects of new drugs, will help in determining the function of these protein complexes.

It is important to note that oscillatory responses in microtransplanted oocytes were variable and dependent on the time of recording as a function of oocyte injection and oocyte extraction. Most oocytes extracted 2 weeks before injection and recorded 24 h after injection failed to show metabotropic responses; therefore, for the optimal generation of metabotropic responses, it is recommended to use oocytes freshly extracted and recorded less than 48 h post-injection.

In contrast to ionotropic GABA and glutamate receptors that are highly correlated with each other [35], metabotropic receptors were more variable in their amplitude and did not correlate very well with GABA responses, preventing the use of GABA responses for normalization purposes. Despite this variability, it was possible to normalize ACPD responses to the response elicited by saturating concentrations of glutamate.

Our pharmacological characterization indicates that mGluR5 receptors are mainly responsible for the oscillations. It is not clear why we did not observe oscillatory responses elicited by mGluR1, which are also present in cortical synaptosomes but with lower abundance. We speculate that ionotropic receptors could have interacted with mGluR1 in an inhibitory manner, possibly by competing with mGluR1 to bind to a specific motif on a scaffold protein. Alternatively, Ca^2^⁺/calmodulin-dependent protein kinase IIα has been shown to effectively inhibit the agonist-induced response of mGluR1, so a similar interaction may have occurred in the microtransplanted synaptosomes [2,47]. Interestingly, oocytes injected with cerebellar synaptosomes which are enriched in mGLuR1, but not mGluR5, showed oscillatory responses. Future studies elsewhere should provide more insights into cerebellar mGluR1 receptors responses and the differences with cortical ones.

To the best of our knowledge, this is the first time that microtransplantation of glutamate metabotropic receptors is reported, extending the growing list of proteins that have been successfully microtransplanted in addition to ionotropic receptors, like chloride channels [48] and voltage gated sodium and calcium channels [49,50,51]. Moreover, this method could be applied to record mGluRs responses from postmortem human brain tissue, although it is still not clear whether long postmortem intervals will be viable for microtransplantation—future studies should be directed to test this possibility.

## Figures and Tables

**Figure 1 membranes-12-00931-f001:**
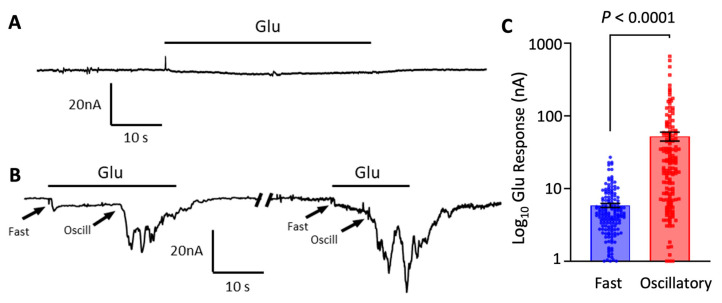
The effects of 1 mM glutamate (Glu) on oocytes microtransplanted with rat cortex synaptosome membranes. Representative two electrodes voltage clamp responses to 1 mM Glu on a non-injected (**A**) and a microtransplanted oocyte (**B**). Recordings were done 24–28 h after injection. A subset of microtransplanted oocytes showed an immediate fast-activating inward current followed by a delayed oscillatory current. Arrows indicate the beginning of the response. Oscillatory currents appeared after a delay within a range of 3–38 s. (**C**) Amplitude of fast and oscillatory Glu responses in microtransplanted oocytes showing both responses (52.2 ± 7.3 nA; *n* = 159 oocytes from 17 frogs; Mean ± SEM). Paired *t*-test.

**Figure 2 membranes-12-00931-f002:**
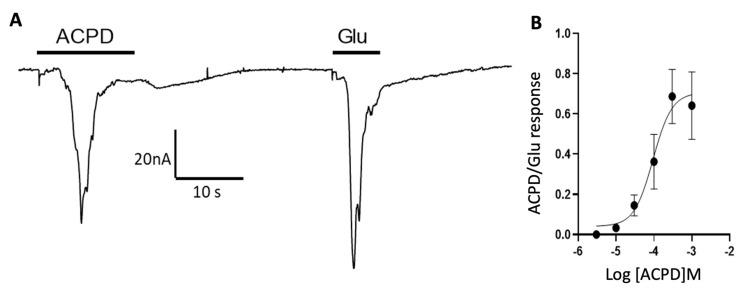
Oscillatory responses are mediated by Group 1 mGluRs. (**A**) Representative ion currents elicited by 100 μM ACPD and 1 mM Glu in the same microtransplanted oocyte. (**B**) Concentration response curve for ACPD normalized by the maximum amplitude of the response elicited by 1 mM Glu in each oocyte (each point represents the mean ± S.E.M from 4 to 13 oocytes tested for each concentration).

**Figure 3 membranes-12-00931-f003:**
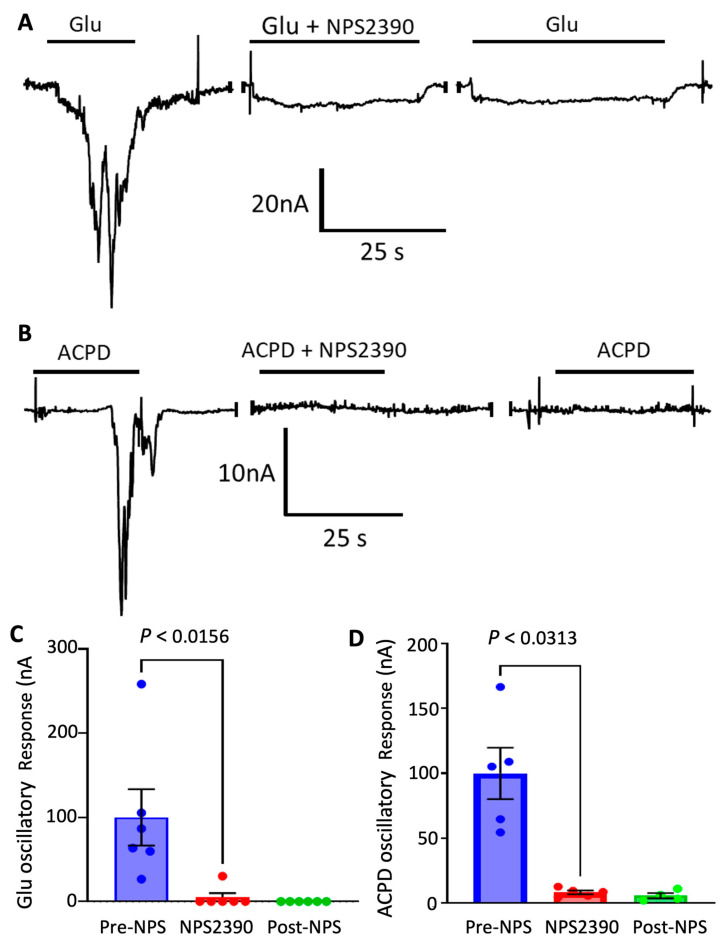
Antagonism of Group 1 receptors inhibits oscillatory glutamate responses. (**A**) Glutamate currents from the same microtransplanted oocyte; each application was separated by a 4 min washing period of Ringer’s solution. A complete loss of oscillations is shown during the application of 10 µM NPS 2390, a mGluR group 1 antagonist (*n* = 6 oocytes from 1 frog). (**B**) ACPD currents of a microtransplanted oocyte with each application type separated by a wash period of 4 min of Ringer’s solution. (**C**,**D**) 10 µM of NPS 2390 co-applied with glutamate or ACPD resulted in a complete loss of oscillations (*n* = 6 and 5 oocytes from 2 frogs, for C and D, respectively).

**Figure 4 membranes-12-00931-f004:**
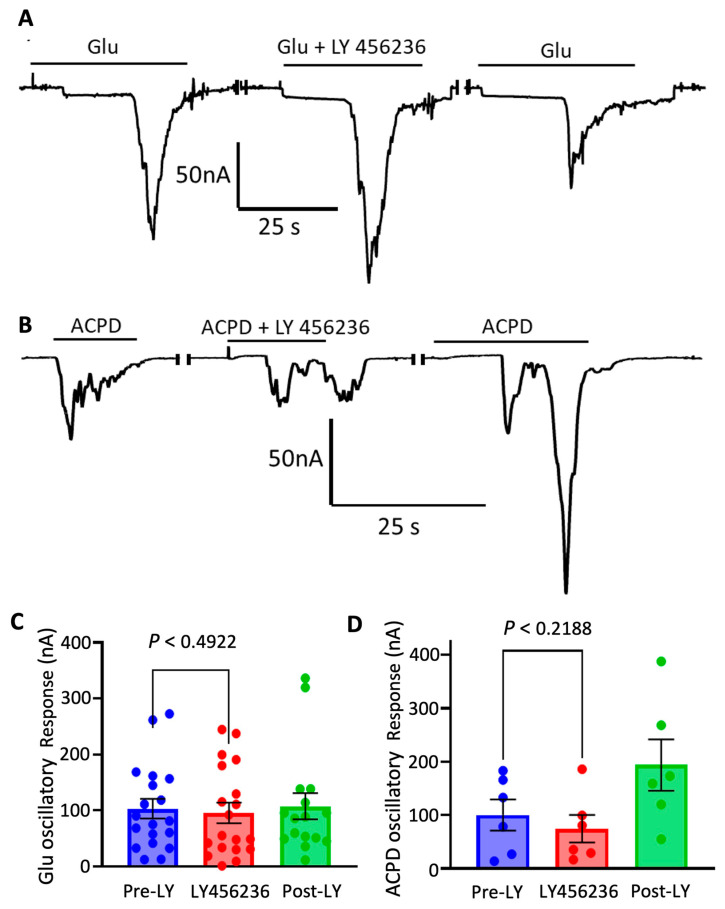
Antagonism of mGluR1 has no effect on metabotropic oscillations. (**A**) Glutamate currents of a microtransplanted oocyte in response to application of the mGluR1 antagonist, LY 456236, with each application separated by 4 min of wash by Ringer’s solution. No significant effect on metabotropic responses was shown (**C**) when LY 456236 was co-applied with glutamate (*n* = 13 oocytes from 4 frogs). (**B**) ACPD currents of a single microtransplanted oocyte in response to application of the mGluR1 antagonist, LY 456236, with each application separated by 4 min of wash by Ringer’s solution. There was no significant change in amplitudes (**D**), although a slight increase in current was observed in some oocytes after a wash period (*n* = 6 oocytes from 2 frogs).

**Figure 5 membranes-12-00931-f005:**
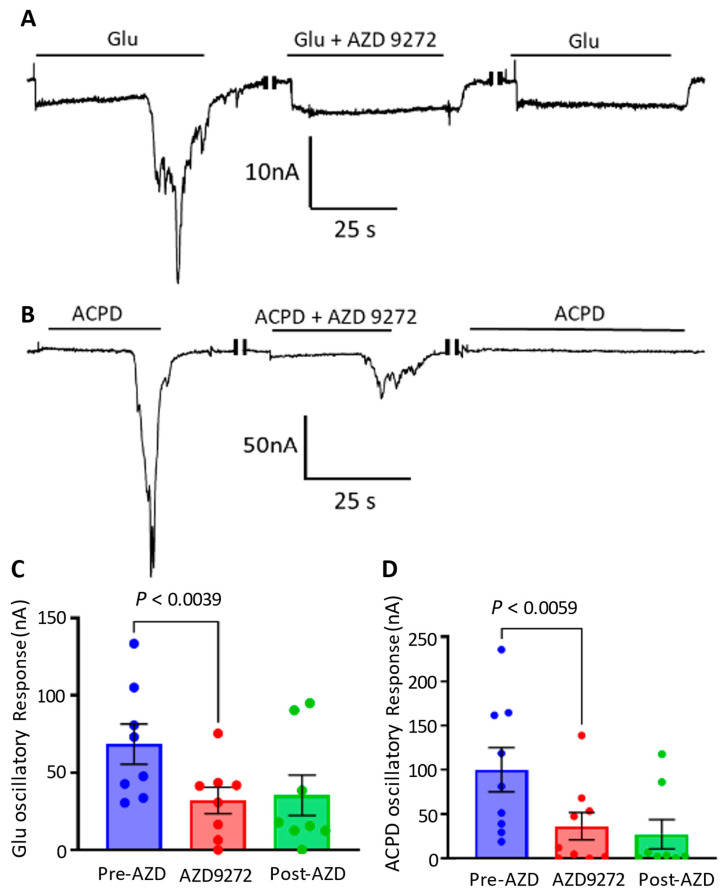
Inhibition of mGluR5 affects presence of metabotropic glutamate oscillations. (**A**) Glutamate currents of a single microtransplanted oocyte in response to application of the mGluR5 antagonist, AZD 9272, with each method of application separated by 4 min of wash by Ringer’s solution. Antagonism of mGluR5 alongside glutamate agonism displayed a significant reduction of responses (**C**), a trend which would prolong into the washing period (*n* = 9 oocytes from 2 frogs). (**B**) ACPD currents of a single microtransplanted oocyte in response to application of the mGluR1 antagonist, AZD 9272, with each method of application separated by 4 min of wash by Ringer’s solution. Blocking of mGluR5 with ACPD (**D**) showed a significant reduction in response that persisted into the post-antagonist application (*n* = 9 oocytes from 2 frogs).

**Figure 6 membranes-12-00931-f006:**
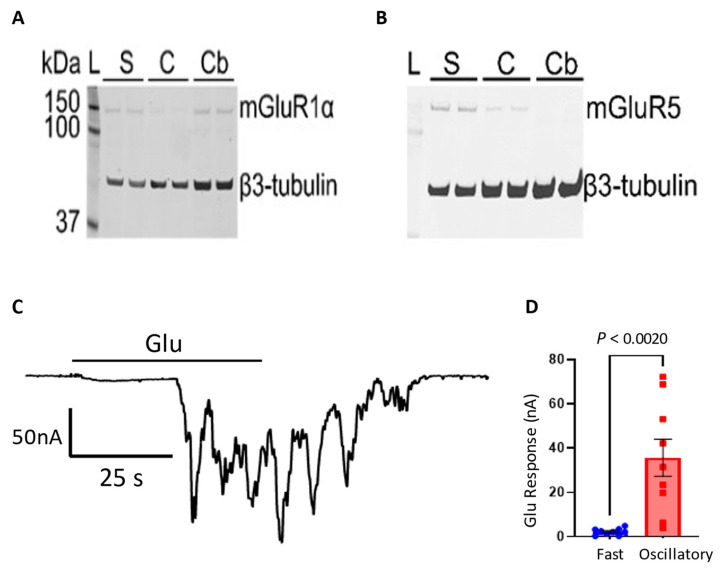
Enrichment of mGluR5 in cortical synaptosomes. Representative immunoblots for mGluR1α (**A**) and mGluR5 (**B**) of lysates from rat cortex synaptosomes (“S”), cortex (“C”) and cerebellum (“Cb”) probed with anti-mouse mGluR1α and anti-rabbit mGluR5 antibodies. Bands for mGluR1α and mGluR5 ran at ∼150 kDa. β3-tubulin (∼50 kDa) was used as loading control. *L*, molecular mass marker. Electrophysiological recordings were made (**C**) 24–28 h after injection using synaptic membranes isolated from rat cerebellum with amplitudes (**D**) of oscillatory Glu responses in microtransplanted oocytes showing responses (35.74 ± 8.38 nA; n = 9 oocytes from 1 frog; Mean + SEM). Paired *t* test.

## Data Availability

The data generated and/or analyzed during the current study are available from the corresponding author on reasonable request.

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
