# Peer review of "Microtransplantation of Postmortem Native Synaptic mGluRs Receptors into Xenopus Oocytes for Their Functional Analysis"

_membranes, 2022, doi:10.3390/membranes12100931_

Round 1

Reviewer 1 Report

Reviewer Comments

In the manuscript: membranes-1880298 titled ‘Microtransplantation of postmortem native synaptic mGluRs receptors into Xenopus oocytes for their functional analysis. ’, Limon and colleagues report a new way, microtransplantation, to introduce native synaptic mGluRs receptors within their native cell membrane environment as a whole unit to the cell membrane of Xenopus oocytes. In this way, the authors can check the activity of mGluRs in a more native state than that through overexpression of mGluRs using plasmids or viruses. This is a brilliant design, and this strategy works well, as demonstrated by the reported results. The experiments were well designed. Their experimental results are physiologically relevant. One of the results is very exciting: the presence of functional group I mGluR in the synaptic membrane and successful coupling to the inositol signal in the oocyte by micro plantation. The method introduced in this manuscript can be easily adapted to other membrane proteins and complexes.

General:

The manuscript was well written. The new approach of microtransplation is crucial for testing the protein function in near-native states in oocytes. This manuscript is significant in terms of its potential wide application for membrane protein functional assay.
Title:
The title looks good
Keywords:
mGluR, Microtransplatation should be included in the Keywords.
Main text:
Well written.
Figures:
Well organized.
Author Contributions:
Information for the author contribution is currently missing. Please provide detailed information.
References:
Please check references to make sure the format is consistent.
Reference 4, 9, 12, 35. The format of these references is inconsistent with other references

Author Response

We thank the reviewers for their comments that helped to improve the manuscript.

Please find attached our point-by-point responses in blue color

R1: In the manuscript: membranes-1880298 titled ‘Microtransplantation of postmortem native synaptic mGluRs receptors into Xenopus oocytes for their functional analysis. ’, Limon and colleagues report a new way, microtransplantation, to introduce native synaptic mGluRs receptors within their native cell membrane environment as a whole unit to the cell membrane of Xenopus oocytes. In this way, the authors can check the activity of mGluRs in a more native state than that through overexpression of mGluRs using plasmids or viruses. This is a brilliant design, and this strategy works well, as demonstrated by the reported results. The experiments were well designed. Their experimental results are physiologically relevant. One of the results is very exciting: the presence of functional group I mGluR in the synaptic membrane and successful coupling to the inositol signal in the oocyte by micro plantation. The method introduced in this manuscript can be easily adapted to other membrane proteins and complexes. The manuscript was well written. The new approach of microtransplation is crucial for testing the protein function in near-native states in oocytes. This manuscript is significant in terms of its potential wide application for membrane protein functional assay.

We thank the reviewer comments

R1: mGluR, Microtransplatation should be included in the Keywords.

Included in current resubmission

R1: Information for the author contribution is currently missing. Please provide detailed information.

Included in current resubmission

R1: Please check references to make sure the format is consistent.
Reference 4, 9, 12, 35. The format of these references is inconsistent with other references

Corrected in current resubmission

Author Response

R2 The manuscript by Miller et al investigates the expression of native glutamate receptors in the Xenopus oocyte model using the microtransplantation approach with synaptosomes isolated from rat cerebral cortex. The main contribution of this paper is the first functional characterization of metabotropic glutamate receptors using the microtransplantation technique, especially mGluR5. This work contributes to expand the repertoire of receptors and channels that can be studied from postmortem tissues using the microtransplantation technique, which could be of primary importance to characterize channel and receptor dysfunctions of patients with neurological disorders in their protein and lipid context.

We thank the reviewer for the comments

The entire manuscript is compelling and well written, the experiments are well conducted and the protocol is clear and well described. Only few minor revisions are required before publication.

  1. Two small additions need to be made in the introduction. The microtransplantation technique must first be described in more detail by briefly recalling the history of its development, the tissues already used, etc.

Thanks for the suggestion, we have included the information requested

  1. Next, a rapid evocation of the macromolecular complex of mGluR receptors could be made, in particular a reminder of the potential protein partners of mGluR5.

Thanks for the suggestion, we have included this information in the introduction and also briefly in the discussion

  1. At the end of the discussion (after L292), it is necessary to mention other ion channels and receptors that have been successfully microtransplanted into the Xenopus oocyte and consider the importance of including metabolotropic receptors to the existing list of microtransplantable ion channels in the physiological and physiopathological context.

Thanks for the suggestion, we have added more information in the discussion.

  1. One of the claim of the manuscript is to record native mGluR5 receptor in their native context (L63-65). This is indeed one of the major advantages of this approach. However no evidence is provided in the manuscript to really claim it. The manuscript would benefit if any or all of the following suggestions could be added to the work:
  • Comparison between microtransplanted mGluR5 and overexpressed mGluR5 (cDNA or RNA injection) on time-dependent oscillatory response decreased (L151). Is the oscillatory response recorded with recombinant mGluR5 is also decreasing after 24h?
  • Comparison between microtransplanted mGluR5 and overexpressed mGluR5 (cDNA or RNA injection) on ACPD/Glu response (L1282-L283). Is the shape of the ACPD/Glu response observed with microtransplanted synaptosomes is modified with overexpressed recombinant mGluR5?
  • Is potential partners of mGluR5 are present in the microtransplanted rat synaptosomes? For example, using western blot approach, identifying cytoplasmic signaling proteins like Fyn would be a convincing indication.

We agree with the reviewer that any of the suggestions above will strengthen the claim that microtrasnplanted receptors are closer to their native context. But the electrophysiological experiments will take a long time to perform and the comparison between microtransplanted vs heterologous receptors is another large study in itself. Therefore, we decided to analyze ours, and other groups, synapto-proteomic data and confirmed that interacting proteins of mGluR receptors are among the most abundant PSD-95 proteins consistently found in synaptosome preparations using the method we also used. This, in our opinion, provides robust evidence that mGluR5 scaffolding interacting proteins are also transplanted into the xenopus oocyte. We have included this information in the introduction.

  1. Finally cortex and cerebellum extracts have been included in the western blot (L219). Is cerebellum and/or cortex extracts have been also microtransplanted? Whether it has been successful or not, it should be mentioned in the manuscript, at least as data not shown.

Yes, Thank you for this comment. We microtransplanted synaptosome preparations and observed oscillatory responses. We have included this data in the manuscript.